# Parallel multi-criteria decision analysis for sub-national prioritization of zoonoses and animal diseases in Africa: The case of Cameroon

Serge Eugene Mpouam[1]*, Dalida Ikoum[1,2], Limane Hadja[1], Jean Pierre Kilekoung Mingoas[1], Claude Saegerman[3]*

1 School of Veterinary Medicine and Science, University of Ngaoundéré, Ngaoundéré, Cameroon, 2 National Program for the Prevention and Fight Against Emerging and Re-emerging Zoonoses, Yaoundé, Cameroon, 3 Faculty of Veterinary Medicine, Unit of Epidemiology and Risk Analysis Applied to Veterinary Science (UREAR-ULiège), Fundamental and Applied Research for Animals & Health (FARAH) Center, University of Liege, Liege, Belgium

* claude.saegerman@uliege.be (CS); sempouam@yahoo.fr (SEM)

**Data Availability Statement:** All relevant data are within the paper and its Supporting information files.

## Abstract

The use of multi-criteria decision analysis (MCDA) for disease prioritization at the sub-national level in sub-Sahara Africa (SSA) is rare. In this research, we contextualized MCDA for parallel prioritization of endemic zoonoses and animal diseases in The Adamawa and North regions of Cameroon. MCDA was associated to categorical principal component analysis (CATPCA), and two-step cluster analysis. Six and seven domains made of 17 and 19 criteria (out of 70) respectively were selected by CATPCA for the prioritization of zoonoses and animal diseases, respectively. The most influencing domains were "public health" for zoonoses and "control and prevention" for animal diseases. Twenty-seven zoonoses and 40 animal diseases were ranked and grouped in three clusters. Sensitivity analysis resulted in high correlation between complete models and reduced models showing the robustness of the simplification processes. The tool used in this study can be applied to prioritize endemic zoonoses and transboundary animal diseases in SSA at the sub-national level and upscaled at the national and regional levels. The relevance of MCDA is high because of its contextualization process and participatory nature enabling better operationalization of disease prioritization outcomes in the context of African countries or other low and middle-income countries.

## 1. Introduction

Low and Middle-Income Countries (LMICs) are significantly and negatively affected by the burden of animal and human infectious diseases [1]. The impact of 35 priority animal diseases is estimated to cost almost USD 9 billion per year in lost productivity in Africa, equivalent to 6% of the total value of the livestock sector in that continent [2]. Zoonotic diseases account for

**Funding:** This work is funded through the Innovative Methods and Metrics for Agriculture and Nutrition Action (IMMANA) programme, led by the London School of Hygiene & Tropical Medicine (LSHTM). IMMANA is co-funded with UK Aid from the UK government and by the Bill & Melinda Gates Foundation INV-002962 / OPP1211308.

**Competing interests:** No.

about 2.5 billion cases of human illness and 2.7 million human deaths worldwide each year [3]. Furthermore, LMICs' capacities to address animal and human health issues are poor partly because of a lack of resources and porous borders despite important human and animal flows between neighboring countries [4]. Livestock movements across porous borders is also responsible of the spread of transboundary animal diseases (TADs) which are diseases that are of significant economic, trade and/or food security importance for several countries in the region [5]. Cameroon is an environmentally diverse country, ranging from tropical rainforest to high mountains and the arid Sahel, resulting in a rich biodiversity [6]. This agroecological diversity, which has greatly contributed to shape Cameroon's production systems, makes this country a good model for the study of zoonoses and animal diseases [6]. Several zoonoses and animal diseases are circulating in Cameroon with various degrees of health and economic importance [7]. The country as many sub-Sahara Africa (SSA) countries is subject to in and out country livestock flows from farming sites to consumption localities or marketplaces [4,7].

The international community, following the emergence and re-emergence of several zoonoses has triggered the mobilization of funds to prioritize transboundary zoonoses in several LMICs [8,9]. Prioritization of diseases is considered as one of the main preliminary steps as far as good emergency management practices of zoonoses and animal diseases are concerned [10,11]. Yet it is a dynamic process that must be contextualized according to several factors including health, economic, socio-cultural and environmental impacts [12]. The methods that are used for prioritization can be qualitative, semi-quantitative and quantitative, all having strengths and weaknesses that need to be taken into consideration during planning steps of the process [12,13]. Disease prioritization has been carried out in several countries in Africa as part of several programs to fight against zoonoses and animal diseases in resource limited contexts [9]. Prioritization of diseases is a decision-making aid tool that enables classification to be made from those diseases which should most hold the attention of public authorities [9] and which should be placed in the forefront in terms of resource allocation for a given prevention or control program. Prioritization of zoonotic diseases has been identified by national representatives as the first step towards managing the public health challenges associated with zoonotic diseases [13]. In Cameroon, as well as several other SSA countries the prioritization of zoonoses at the national level was done using the tool developed by the Center for Diseases Control and prevention (CDC) "One Health Zoonotic Disease Prioritization tool" or the OIE "Phylum tool" [8,9,13,14]. Even though those processes are also meant to be applied at different scales they mainly involved decisions makers or experts from central or national levels with very few participants from the local level and private sector [9]. Furthermore, some countries only considered zoonoses and did not prioritize animal diseases with no human health impact. For a better operationalization of disease prioritization results, a holistic and more participatory approach concerning both zoonoses and animal diseases should be implemented to make sure that the interests of decision makers match with the ones of farming communities [15]. Such prioritization at the local level would allow better adhesion of stakeholders, better targeting of priority zoonoses and animal diseases while further reducing prevention and control costs through adapted preparedness and response capacities. However, prioritization processes implemented in SSA countries to date were too elitist and lightly involved the private sector and field practitioners [9]. This might result in different diseases impact perception and therefore priorities between central or national experts and locals (field practitioners and farmers) [9]. Recent trends advocate for the use of transparent, reproducible, and inclusive methods using multiple criteria organized in different groups or categories [12,13,16]. Though poorly used in the context of animal diseases, in 2019, a multicriteria and cost-effective ranking method was applied in Belgium to prioritize the emergence of exotic diseases of animals from a list of selected diseases based on the opinion of experts and a survey on drivers of emergence

related criteria [17]. Yet, in SSA livestock production system types vary from intensive to extensive [18] and the epidemiological situation is characterized by the presence of a range of endemic diseases associated to the continuous threats of emerging or reemerging zoonoses or animal diseases [7].

In this study, we conducted a parallel prioritization of endemic zoonoses and TADs using multi-criteria decision analysis (MCDA) at the local level in other to design a framework for the contextualization of the process enabling better operationalization of disease prioritization outcomes in the context of African countries or other LMICs.

## 2. Materials and methods

### 2.1. Study site

This study was carried out from September 2020 until February 2021. Diseases prioritized in the context of the present study concerned two neighbouring areas: the Adamawa and North regions of Cameroon. These regions are in the high guinea savanna and sudano-sahalian agro-ecological zones. In the high guinea savanna, the mean annual temperature is 19.4°C and in the sudano-sahalian zone temperatures are on average 40–42°C and 17°C for the highest and lowest [6]. Annual precipitation cumulates to 2000 mm and 900 mm in the Adamawa and the North region respectively [6]. The Adamawa region is a pastoral highland of approximately 64,000km$^2$ and 45% of the North region is dedicated to agriculture and reserve zones respectively [19]. The Adamawa and the north regions are the main livestock production areas of Cameroon, with an official cattle population of about 1,250,000 [20] and 1,800000 [19] respectively. Both regions are also transhumance areas for herds coming from other areas of Cameroon during the dry season (October—April). In both regions, various livestock species such as cattle, small ruminants, pigs, poultry are reared by local communities [6] but a recent census or estimates of farm animals in these regions is yet to be published. For the purpose of this study, all those species reared in the study area were considered as farm animals.

### 2.2. Identification of zoonoses and animal diseases

We have established an exhaustive list of zoonoses and animal diseases or infections (bacterial, parasitic, and viral) circulating in Cameroonian and neighbouring countries from various data sources. Thus, the database was created using publications, health information systems, World Health Organization notifications for Cameroon, Country diseases notifications to EMPRES (of the United Nations Food and Agricultural Organization) or WAHIS (of the World Organization for Animal Health), disease reports from the Ministry of Livestock, Fisheries and Animal Industries, the Ministry of Public Health and the Ministry of Forests and Wildlife at central and local levels. Online search of diseases was done on several websites including Google Scholar, PubMed, Science Direct, PLOS ONE, Online report, Medline, African Journal Online. The disease duplicates were systematically removed. In total, 534 sources were used from Cameroon and its neighbouring countries with various share according to disease types (Table 1).

**Table 1. Number of sources used to create disease lists.**

| | Number of sources by disease type | | | |
|---|---|---|---|---|
| | **Bacterial disease** | **Parasitic disease** | **Viral disease** | **Total** |
| Cameroon [7] | 39 | 94 | 36 | 139 |
| Neighboring countries (Unpublished data) | 181 | 128 | 86 | 395 |
| Total | 220 | 222 | 122 | 534 |

**Table 2. Inclusion and exclusion criteria used to identify zoonoses and animal diseases to be prioritized.**

| Criteria | |
|---|---|
| **List of zoonoses** | **List of animal diseases** |
| a) Being a zoonotic disease | i) Being a disease of farm animals |
| b) Notified in animals at least once in both regions during the last four years | j) Studied in Cameroon the last 20 years (published and unpublished studies) |
| c) Notified in humans at least once during the last four years | k) Present in neighboring countries and notified to the WAHIS during the last 20 years |
| d) Zoonosis with known significant socio-ecological impact | l) Present in both studied regions during the study period |
| e) Having a known socio-economic impact | |
| f) Zoonosis present in at least two agroecological zones and in border countries. | |

## 2.3. Selection of zoonoses and animal diseases

A set of inclusion and exclusion criteria were used with respect to prioritization objectives subject to prior validation with the Cameroon animal disease surveillance network (RESCAM). These exclusion and inclusion criteria were applied for the selection of diseases of particular interest at the local level. Thus, upon completion of the database, diseases were excluded if they did not meet the criteria provided on Table 2 (top-down order, a) to f) for zoonoses and i) to l) for animal diseases) and following the inclusion-exclusion algorithm presented on Fig 1. The final list was validated with the animal disease surveillance network of the country.

## 2.4. Prioritization criteria: Questionnaire design

Two set of questionnaires or list of criteria were used: a first preliminary list of 70 criteria with their indicators and measurement scales grouped into seven disease impact domains (epidemiology-infectiology, public health, disease control, economy and trade, society, livestock production system, society, and environment) was used to create a final and second short list of prioritization criteria. The 70 criteria were submitted to experts to assess their relevance and importance.

## 2.5. Expert's prior knowledge

Two types of experts were used (Table 3). Firstly, a group of experts called for the purpose of this study criteria experts (S1 and S2 Tables) who evaluated all 70 criteria, and selected zoonosis and animal disease final prioritization criteria to be included in the short list based on their relevance and importance. Secondly another group of experts namely disease experts (S3 and S4 Tables) who scored the criteria relevance and importance for each selected disease (each disease expert gave an opinion on a maximum of six diseases and each disease got three to four different expert opinions).

## 2.6. Data collection

Two rounds of surveys were carried out. For the first round, criteria experts were contacted by email and telephone for information and consent purpose. Data collection sheets were also shared and based on expert convenience, data collection was carried out online or through face-to-face interview. The latter was the method used in the second survey involving disease experts at local level.

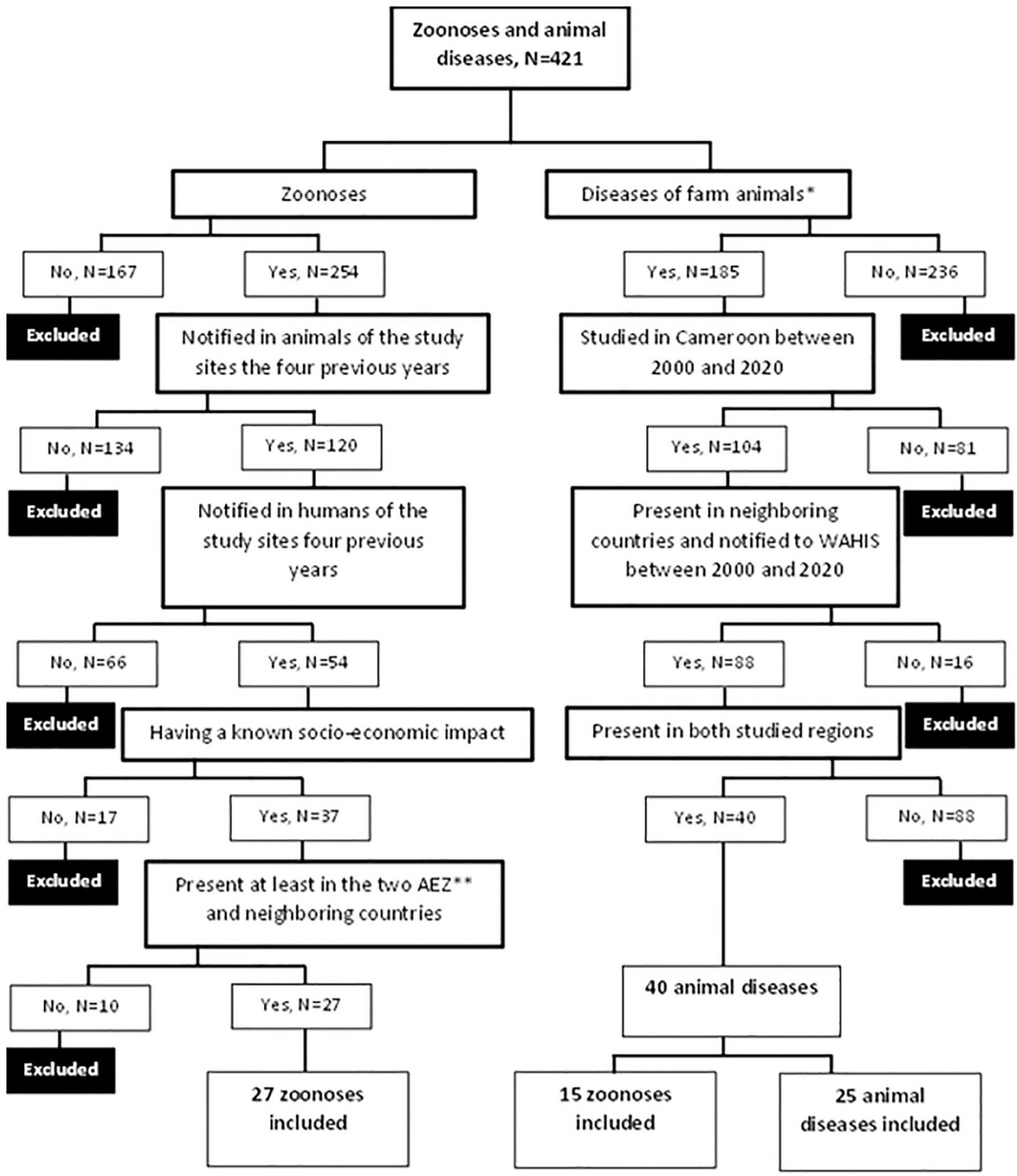

**Fig 1. Systematic process for selecting the zoonoses and animal diseases.** *farm animals: Cattle, sheep, goats, swine, and poultry; ** AEZ: Agro-ecological zone.

**Table 3. Number of experts used for the parallel prioritization of zoonoses and animal diseases.**

| Prioritization | Number of Criteria experts | Number of disease experts | Total |
|---|---|---|---|
| Zoonoses | 15 | 25 | 40 |
| Animal diseases | 21 | 40 | 61 |

## 3. Calculation and data analysis

Like data collection, data was analysed in two rounds.

### 3.1. First round of data analysis

The relevance score and importance weight of criteria collected through the survey of criteria experts were subject to normalization prior to data analysis. The criteria weight was then calculated by multiplying the normalized relevance score with intradomain weight of the importance to reflect the relative preference of experts on each criterion. In order to identify final prioritization criteria, criteria weights were subject to categorical principal component analysis (CATPCA), a nonlinear principal component analysis (PCA) available in the Categories Module of SPSS® [21,22] that can handle multiple variables of various types (nominal, ordinal, and numerical) simultaneously, and can deal with nonlinear relationships between variables or criteria [22]. Criteria were tested for correlations. Spearman correlation test (correlation coefficient of $\geq 0.7$ as benchmark) was used to assess the independence of criteria. The significance level for correlations was set at $p < 0.01$) [23]. The exclusion of strongly correlated criteria (with lowest loading score) associated to the selection of criteria with a loading score $\geq 0.5$ on one of the two components set, finally resulted in the identification of the most discriminating criteria that were therefore used as disease prioritization criteria (Table 4). Similar cut-off levels have been used by several authors for significant indicator loadings [18,23–25]. Thus, higher the loading of a criteria on a given principal component (or dimension), the more that variable contributes to the variation accounted for by this component.

### 3.2. Second round of data analysis

**Calculation of disease scores:** In the second round, the score (relevance) and intra/interdomain weight (importance) collected for each disease were also subject to normalization. To obtain the overall weight score calculated by expert/disease and the final mean score used for the ranking of diseases, an aggregation method that combined the two types of weighting (intra/interdomain) was used according to the methodology already described [17]. Higher the overall weight score, the more the disease is considered as priority according to experts. By using the median, a ranking was made to assess whether there was a difference with the ranking obtained using the mean. The ranges were used to assess which diseases had the highest and lowest levels of variation of the mean score or prioritization uncertainty among disease experts.

**Classification of domains:** The domains were ranked to determine which domains were considered the most influential for the prioritization of zoonoses and animal diseases. Domain ranking was performed using cross-domain scores (weights). The sum of each domain weight by disease given by each expert was classified (top-down position). Then, domain ranking was displayed as the frequency at which each domain was ranked for a given position.

**Sensitivity analysis:** Two sensitivity analyses were carried out that is one on the influence of domains and the other one on the influence of expert groups. The sensitivity analysis on domains was performed by deleting one domain and recalculating the mean scores to classify

**Table 4. Criteria selected (in bold) for prioritization of zoonoses and animal diseases through CATPCA (domain, criteria and their CATPCA loading of components on the two dimensions for zoonoses and animal diseases).**

| Domain | Criteria | Dimension | | | |
| --- | --- | --- | --- | --- | --- |
| | | Zoonoses | | animal diseases | |
| | | 1 | 2 | 1 | 2 |
| Epidemiology | Morbidity rate (%) | 0.254 | -0.531 | -0.018 | 0.323 |
| | **Mortality rate (%)** | **0.835** | -0.274 | **0.675** | **0.669** |
| | **Lethality rate (%)** | **0.831** | -0.295 | **0.729** | **0.645** |
| | Prevalence (%) | -0.559 | -0.153 | -0.435 | 0.049 |
| | Pathogeny | -0.229 | 0.405 | -0.496 | 0.141 |
| | Genetic variability of the infectious agent | -0.321 | 0.306 | -0.325 | 0.094 |
| | Agent specificity | -0.637 | -0.110 | -0.334 | 0.178 |
| | Transmission mode | -0.435 | 0.095 | -0.529 | 0.260 |
| | Incubation period | -0.313 | 0.437 | -0.588 | 0.159 |
| | Clinical course | 0.052 | 0.303 | -0.285 | 0.417 |
| | **Environmental Persistence** | -0.210 | **0.684** | 0.247 | -0.186 |
| | Endemic potential | -0.204 | 0.466 | -0.404 | 0.274 |
| | **Evolutive character of the pathogen agent** | -0.246 | **0.710** | 0.239 | -0.117 |
| | Species | | | | |
| | Presence/absence of vector(s) and/or reservoir(s) in Cameroon or in transboundry countries | -0.518 | -0.059 | -0.675 | 0.322 |
| Control—prevention | **Control of reservoir(s) and/or vector(s)** | -0.169 | 0.137 | -0.286 | **0.600** |
| | **Vaccination** | **0.837** | -0.169 | **0.634** | **0.702** |
| | **Treatment** | **0.837** | -0.169 | **0.679** | **0.648** |
| | Availability and quality of diagnostic tools | -0.558 | -0.053 | -0.754 | -0.614 |
| | Knowledge on the pathogen agent | -0.219 | -0.188 | -0.612 | 0.238 |
| | Effectiveness of control measures (other than treatment, vaccination and vector(s)/reservoir(s) control) | -0.318 | -0.504 | -0.646 | 0.332 |
| | Effectiveness of prevention (other than vaccination) | -0.381 | -0.559 | -0.681 | 0.432 |
| | **Surveillance of the pathogen agent** | -0.258 | **0.738** | -0.703 | **0.232** |
| | Detection of emergence—for example difficulties for the farmer/veterinarian or patient to declare the disease or clinical signs not so evident. | 0.085 | -0.580 | -0.358 | 0.558 |
| | Antibiotic resistance | -0.311 | 0.350 | -0.401 | 0.350 |
| | Transport vehicles of live animals | -0.176 | -0.119 | -0.547 | 0.310 |
| | Efficiency of the local veterinary network | -0.496 | -0.150 | -0.671 | 0.126 |
| Economy-trade | **Production losses (milk, eggs, growth, meat)** | **0.837** | -0.169 | **0.672** | **0.694** |
| | Mandatory slaughtering | -0.242 | -0.372 | 0.336 | 0.438 |
| | **Additional costs: treatment, vaccination, disinfection, labor** | -0.258 | -0.382 | -0.093 | **0.689** |
| | Limitation of importation-exportation | -0.630 | -0.006 | -0.597 | 0.171 |
| | Disturbance of demand and supply (fall in prices, etc) | 0.029 | -0.741 | -0.497 | 0.422 |
| | Impact on related sectors (tourism, animal feeds, etc.) | -0.647 | 0.372 | -0.693 | 0.320 |
| | **Impact on cattle industry** | -0.478 | -0.601 | -0.453 | **0.589** |
| | Impact on small ruminants' industry | -0.513 | -0.500 | -0.492 | 0.230 |
| | Impact on swine industry | -0.661 | -0.433 | -0.597 | 0.192 |
| | Impact horse industry | -0.634 | -0.438 | -0.437 | 0.425 |
| | Impact on poultry industry | -0.746 | -0.300 | -0.713 | 0.322 |
| | Impact on wildlife industry | -0.485 | -0.730 | -0.626 | 0.048 |
| | Zoonotic impact (costs of treatment per person—cost of illness) | -0.518 | -0.631 | -0.748 | 0.036 |
| | Zoonotic impact (costs of prevention per person) | -0.652 | -0.263 | -0.809 | 0.048 |
| | Economic impact (cost of treatment—cost of illness) | -0.475 | -0.426 | -0.736 | 0.151 |
| | Impact per animal (prevention costs) | -0.545 | -0.650 | -0.744 | 0.249 |

*(Continued)*

**Table 4.** (Continued)

| Domain | Criteria | Zoonoses | | animal diseases | |
|---|---|---|---|---|---|
| | | 1 | 2 | 1 | 2 |
| | Zoonotic / common** agent | -0.375 | -0.411 | -0.514 | -0.392 |
| Public health | **Classification of zoonoses** | -0.096 | **0.757** | -0.507 | 0.329 |
| | **Disease knowledge in humans** | -0.212 | **0.701** | -0.232 | 0.624 |
| | **Morbidity rate (%)** | **0.845** | -0.345 | **0.660** | 0.521 |
| | **Mortality rate (%)** | **0.815** | -0.364 | **0.682** | **0.655** |
| | **Lethality rate (%)** | **0.837** | -0.271 | **0.674** | **0.655** |
| | Contamination route | -0.446 | -0.605 | -0.431 | -0.257 |
| | After-effects or negative impact on the patients' quality of life | -0.690 | -0.361 | -0.730 | 0.251 |
| | Presence of a fight plan (vaccination scheme, determination of populations at risk, surveillance of the disease in humans, definition of areas at risk) | -0.795 | 0.467 | -0.691 | -0.457 |
| | Epidemic potential | -0.290 | -0.675 | -0.113 | -0.560 |
| | Vaccination | -0.770 | 0.394 | -0.535 | 0.164 |
| | Treatment | -0.875 | 0.232 | -0.692 | 0.008 |
| | Availability and quality of diagnostic tools | -0.566 | -0.321 | -0.502 | -0.094 |
| Environment | Agro- ecological zone of predilection for the pathogen or disease | -0.352 | 0.179 | -0.778 | -0.005 |
| | Climatic conditions favoring the occurrence of the disease or pathogen | 0.095 | -0.593 | -0.427 | 0.249 |
| | Influence of annual rainfall in the survival and transmission of the pathogen/disease | -0.046 | -0.092 | -0.174 | 0.267 |
| | Influence of annual humidity in the survival and transmission of the pathogen/disease | -0.250 | 0.038 | -0.094 | 0.381 |
| | Influence of annual temperature in the survival and transmission of the pathogen/disease | -0.565 | -0.122 | -0.541 | 0.175 |
| | **Proximity of the livestock farm to wildlife and disease reservoirs** for example contact with birds and wild or feral animals that searched landfill sites containing contaminated animal products | **0.667** | -0.474 | **0.687** | **0.643** |
| | **Potential roles of protected areas in the emergence of the pathogen** | 0.239 | -0.684 | -0.143 | **0.607** |
| | Proximity of domestic animals and humans to wildlife protected areas and wildlife reservoirs of disease. | -0.200 | 0.675 | 0.044 | 0.490 |
| | **Hunting Activities: hunted animals can be brought back to where livestock is present** | -0.302 | 0.466 | -0.260 | **0.664** |
| Livestock production system | Mono species farms—One single farmed animal or multi species farms (farms with more than one species, for example goats and bovines in the same farm/land/premises) | -0.518 | 0.379 | -0.593 | 0.216 |
| | Transmission of the agent in relation of the possible spread of the epidemic (i.e. ease/speed of spread) | 0.288 | -0.666 | 0.107 | -0.191 |
| | **Farm demography/management: such as type of production-reproduction-fattening-finishing.** | 0.048 | -0.212 | -0.129 | **0.547** |
| | Animal density of farms | -0.414 | -0.638 | -0.468 | 0.096 |
| | Human movements among premises-Veterinarians or farm staff | 0.325 | -0.351 | -0.052 | 0.307 |
| | Feeding practices of farms | 0.406 | -0.401 | -0.118 | **0.523** |
| Society | **Lowered consumption** | **0.658** | 0.243 | 0.089 | 0.383 |
| | Perception of the problem by the consumer (problem poorly known or not known at all, poorly or not controllable at all, affects a sensitive public) | -0.081 | -0.462 | -0.462 | -0.293 |
| | Potential impact of media | 0.446 | -0.245 | 0.307 | -0.136 |
| | **Impact on animal welfare and biodiversity** | **0.852** | -0.173 | **0.674** | **0.661** |

**Bold** for the loading score > 0.5 for at least one of the two dimensions of the CATPCA analysis and was selected among final disease prioritization criteria.

the diseases. This reduced ranking was then compared to the complete model. If the ranking position changed less than three ranks, the final model was considered robust. If it changed three ranks or more, then the domain was considered as influencing disease ranking. In addition, a bivariate Spearman rank correlation between the complete model and each reduced model (models where one domain was not considered for ranking) was used to assess the statistical dependence between the ranking of the two models that is how well the relationship

between both can be described with a monotonic function or trend (varying in the same and unique direction). Thus, a high Spearman correlation (close to 1) meant the higher a disease ranked on the complete model, the higher it ranked on the reduced model as well and vice versa. Likewise, sensitivity analysis also concerned experts belonging to four groups and mean scores for ranking in reduced models were calculated by removing one group of experts. Each reduced disease ranking model was then compared to the complete model and the Spearman rank correlation calculated as well for the same purpose. The significance level was set at $p < 0.05$.

**Cluster analysis**: In order to reveal natural groupings (or clusters) of the ranked diseases, a two-step cluster analysis was performed with SPSS software version 26.0. The overall weight score of diseases of complete and reduced models was used as input for cluster analysis. Several cluster solutions (number of clusters and clusters' composition) were obtained. The selection of the final cluster solution was based on the cluster quality provided by the silhouette measure of cohesion and separation [26]. It is a measure (or a value) of how well an object matches or fits to its own cluster (cohesion) in comparison to other clusters (separation). It values range between -1 and 1 such that a high value indicates that an object matches well to its own cluster and poorly matches to other neighbouring clusters [27,28].

## 4. Results

### 4.1. Disease selection

After applying inclusion and exclusion criteria we compiled two lists: one composed of 27 zoonoses and the other with 40 animal diseases (Table 5). For the prioritization of zoonoses, the list accounted eleven bacterial, eleven parasitic and five viral zoonoses. For the list of animal diseases, 15, ten and 15 were due to bacterial, parasitic, and viral agents respectively. Six out of the 40 diseases selected on the list of animal diseases were zoonoses specific to that list. Nine zoonoses were common to both lists. Apart for human (affected by zoonoses) other animal species could be affected by the listed diseases specifically (Table 5).

### 4.2. Prioritization domains and criteria

In general, criteria experts all agreed with criteria submitted to them and no criteria was deleted apart from minor comments. For the prioritization of zoonoses, one domain (livestock production system) was systematically excluded by categorical principal component analysis (CATPCA). In total from the initial 70 criteria, 17 and 19 prioritization criteria were identified using CATPCA for zoonoses and animal diseases, respectively (Table 4). Ten criteria were common to both processes. For the prioritization of zoonoses only criteria from six domains remained. Criteria from all the seven domains were represented for the prioritization of animal diseases. "Epidemiology and infectiology" and "public health" domains represented more than half of the criteria selected for zoonotic diseases. For the prioritization of animal diseases, control–prevention and public health domains had more than two fifth of all selected criteria.

By using two-step cluster analysis, three clusters were identified for zoonoses and animal diseases (Tables 6 and 7). Inputs used for two-step cluster analysis were the disease mean scores of the complete model (with all domains) and reduced models (where one domain was excluded from ranking). For zoonoses, the clusters were composed of eight, nine, ten diseases and were characterized as "very high importance"," high importance", and "moderate importance". Zoonoses belonging to the "very high importance" cluster were COVID-19, anthrax, filariasis, brucellosis, toxocariasis, bovine tuberculosis, salmonellosis rabies. The cluster of low importance included scabies, streptococcosis, giardiasis, dermatophytosis, cysticercosis, hydatidosis, chlamydiosis, paratuberculosis, swine influenza and echinococcosis. For animal

**Table 5. List of selected zoonoses and animal diseases subject to prioritization.**

| Prioritization of zoonoses (N = 27) | | Prioritization of animal diseases (N = 40**) | |
|---|---|---|---|
| **Zoonosis** | *Causative agent* | **Animal disease** | *Causative agent* |
| Bacterial disease | | Bacterial disease | |
| 1. Anthrax** | *Bacillus anthracis*[C, G, S, O] | 1. Campylobacteriosis | Campylobacter spp[C, G, S] |
| 2. Brucellosis** | *Brucella abortus, Brucella mellitensis*[C, G, S] | 2. Colibaccilosis | *Escherichia coli*[C, G, S, B] |
| 3. Chlamydiosis | *Chlamydia abortus*[G, S] | 3. Contagious bovine pleuropneumonia | *Mycoplasma mycoides susp. mycoides*[C] |
| 4. Clostridiosis | *Clostridium perfringens*[C, G, S, O] | 4. Contagious caprine pleuropneumonia | *Mycoplasma capricolum susp. capripneumonia*[G] |
| 5. Infectious mastitis** | *Staphylococcus aureus, Streptococcus agalactiae*[C] | 5. Dermatophilosis | *Dermatophilus congolense*[C] |
| 6. Listeriosis | *Listeria monocytogenes*[P,] | 6. Heartwater disease | *Ehrlichia ruminantium* |
| 7. Rickettsiosis | *Rickettsia spp*[O] | 7. Infectious endometritis | Various bacteria[C] |
| 8. Paratuberculosis | *Mycobacterium avium subsp. paratuberculosis*[C, G, S, O] | 8. Leptospirosis* | *Leptospira spp*[C,P] |
| Prioritization of zoonoses | | Prioritization of animal diseases | |
| Zoonosis | *Causative agent* | Animal disease | *Causative agent* |
| 9. Salmonellosis** | *Salmonella Typhimurium, Salmonella enteritidis, Salmonella infantis*[C, G, S, O] | 9. Pasteurellosis | *Pasteurella multocida*[B] |
| 10. Streptococcosis | *Streptococcus spp*[C, G, S, O] | 10. Q-fever* | *Coxiella burnetii*[B] |
| 11. Bovine tuberculosis** | *Mycobacterium bovis*[C, G, S, O] | Parasitic disease | |
| Parasitic disease | | 11. Anaplamosis | *Anaplasma spp*[C, G, S, O] |
| 12. Aspergillosis | *Aspergillus spp*[C] | 12. Animal Trypanosomiasis | *Trypanosoma spp*[C, G, S, O] |
| 13. Cryptosporidiosis | *Cryptosporidium spp*[C] | 13. Babesiosis | *Babesia spp*[C, G, S, O] |
| 14. Cysticercosis | *Taenia solium'* | 14. Coccidiosis | *Eimeria spp*[C, G, S, O] |
| 15. Dermatophytosis | *Microsporum spp, Trichophyton spp* | 15. Fascioliasis/ Distomatosis* | *Fasciola hepatica, Fasciola gigantica*[C, G, S] |
| 16. Echinococcosis | *Echinococcus granulosus*[C, S] | 16. Monieziasis | *Moniezia expansa*[C] |
| 17. Hydatidosis | *Taenia hydatigena*[C, G, S, O] | 17. Nematodiasis | *Nematodes (several species)*[C, G, S, O] |
| 18. Esophagostomosis** | *Oesophagostomum spp*[C, G, S,O] | 18. Paramphistomosis | *Paramphistomum spp*[C] |
| 19. Filariasis | *Microfilaria spp*[O] | Viral disease | |
| 20. Giardiasis | *Giardia intestinalis, Giardia duodenalis*[C, O] | 19. African swine fever | African swine fever virus[P] |
| 21. Scabies** | *Sarcoptes scabiei*[C, G, S, O] | 20. Bovine viral diarrhea | Bovine viral diarrhea virus[C] |
| 22. Toxocariasis | *Toxocara canis, Toxocara catis*[O] | 21. Classical swine fever | Classical swine fever virus[P] |
| Viral disease | | 22. Foot and mouth disease | Aphtovirus (FMDV)[C, G, S, P] |
| 23. COVID-19** | *Coronavirus*[O] *(SARS-CoV-2)* | 23. Hokovirosis | Porcine Hokovirus[P] |
| 24. Herpes infections | *Herpes simplex virus*[O] | 24. Infectious bovine rhinotracheitis | Infectious bovine rhinotracheitis virus[C] |
| 25. High pathogenic avian influenza** | High pathogenic avian influenza *virus*[B] | 25. Infectious bursal disease | Infectious bursal disease virus[B] |
| 26. Rabies | *Lyssavirus Rhabdoviridae*[C, G, S, P, O] | 26. Low pathogenic avian influenza* | Low pathogenic avian influenza virus[B] |
| 27. Swine influenza | *Swine Influenza A virus*[P] | 27. Lumpy skin disease | Lumpy skin disease virus[C] |
| | | 28. Newcastle disease | Newcastle disease virus 1[B] |
| | | 29. Rift Valley fever* | Rift Valley fever virus[C, G, S] |
| | | 30. Small ruminant plague | Peste des petits ruminants virus[G, S] |
| | | 31. Viral hepatitis E* | Hepatitis E virus[P] |

* Zoonoses selected and specific to the list of animal diseases,

** Zoonoses (nine) selected in both lists but not added to the animal diseases' list part on this table,

[C, G, S, P, B, O] superscript letters for affected species (except humans for zoonoses); C: Cattle, G: Goat, S: Sheep, P: Pig, B: Bird/poultry, O: Other (dog, cat, apes, monkeys).

**Table 6. Ranking and two-step cluster analysis of 27 zoonoses according to the completed and reduced domain models.**

| Zoonosis | Cluster | Reduced domain | | | | | | |
|---|---|---|---|---|---|---|---|---|
| | | Complete model | Epidemiology and infectiology | Control and prevention | Economy and trade | Public health | Environment | Society |
| | | Mean score | Mean score | Mean score | Mean score | Mean score | Mean score | Mean score |
| | | Rank | Rank | Rank | Rank | Rank | Rank | Rank |
| Anthrax | 1 | 112.86 2 | 83.69 1 | 75.68 14* | 108.49 3 | 81.20 1 | 106.69 3 | 108.51 2 |
| Aspergillosis | 2 | 99.43 10 | 71.23 10 | 79.95 7* | 95.11 10 | 68.01 15* | 90.97 11 | 91.91 11 |
| Brucellosis | 1 | 108.62 4 | 78.52 6 | 81.85 5 | 103.16 6 | 68.85 15* | 105.83 4 | 104.88 3 |
| Chlamydiosis | 3 | 83.35 25 | 55.18 26 | 71.40 19* | 79.09 26 | 57.60 25 | 75.51 26 | 77.98 24 |
| Clostridiosis | 2 | 96.57 15 | 68.32 19 | 75.30 18 | 91.15 17 | 68.68 16 | 89.88 17 | 89.53 16 |
| COVID-19 | 1 | 114.88 1 | 77.78 3 | 95.79 1 | 110.67 1 | 70.09 10* | 109.66 1 | 110.38 1 |
| Cryptosporidiosis | 2 | 97.40 13 | 70.61 12 | 66.54 24* | 93.82 13 | 70.88 8* | 92.84 10 | 92.29 10 |
| Cysticercosis | 3 | 87.65 22 | 65.95 18* | 70.10 21 | 84.04 22 | 53.30 27* | 83.08 19 | 81.77 20 |
| Dermatophytosis | 3 | 88.31 21 | 54.98 27* | 76.94 11* | 84.25 21 | 64.61 17* | 82.76 21 | 78.01 23 |
| Echinococcosis | 3 | 83.44 35 | 63.41 29* | 64.36 33 | 79.02 36 | 60.59 30* | 77.47 32 | 72.33 36 |
| Filariasis | 1 | 111.99 3 | 83.28 3 | 80.52 6* | 109.55 2 | 74.29 6* | 107.96 2 | 104.33 4 |
| Giardiasis | 3 | 89.58 20 | 65.06 20 | 76.37 13* | 84.49 20 | 57.99 24* | 79.51 23* | 84.50 19 |
| High pathogenic Avian influenza | 2 | 100.91 9 | 71.82 9 | 87.83 2* | 96.41 9 | 59.942 23* | 93.64 9 | 94.92 9 |
| Herpes infections | 2 | 97.36 14 | 64.20 21* | 73.81 16 | 94.18 12 | 74.40 5* | 90.07 14 | 90.16 13 |
| Hydatidosis | 3 | 86.96 23 | 65.60 19* | 72.17 17* | 79.94 23 | 56.16 26* | 80.64 22 | 80.30 22 |
| Infectious mastitis | 2 | 97.94 11 | 68.28 17* | 71.59 18* | 92.86 14* | 78.62 2* | 90.65 12 | 87.71 15* |
| Listeriosis | 2 | 95.38 16 | 70.32 13* | 83.80 3* | 89.42 16 | 63.70 18 | 83.76 18 | 85.90 17 |
| Oesophagostomosis | 2 | 95.35 17 | 60.84 25* | 76.82 12* | 88.99 17 | 74.96 3* | 89.17 16 | 85.96 16 |
| Paratuberculosis | 3 | 84.44 24 | 68.95 15* | 54.58 27* | 79.40 25 | 66.39 16* | 76.55 25 | 76.31 25 |
| Rabies | 1 | 103.24 8 | 72.43 8 | 77.84 10 | 100.68 8 | 68.60 13* | 97.44 8 | 99.23 7 |
| Rickettsiosis | 2 | 97.85 12 | 69.13 14 | 70.39 20* | 94.35 11 | 72.90 7* | 90.63 13 | 91.83 12 |
| Salmonellosis | 1 | 104.13 7 | 76.31 7 | 78.01 9 | 100.82 7 | 70.80 9 | 97.88 7 | 96.81 8 |
| Scabies | 3 | 91.46 18 | 63.63 22* | 78.34 8* | 87.76 18 | 63.27 19 | 82.91 20 | 81.40 21* |
| Streptococcosis | 3 | 91.22 19 | 70.89 11* | 67.68 23* | 87.16 19 | 61.54 20 | 84.04 17 | 84.81 18 |

*(Continued)*

**Table 6.** (Continued)

| Zoonosis | Cluster | Reduced domain | | | | | | |
|---|---|---|---|---|---|---|---|---|
| | | Complete model | Epidemiology and infectiology | Control and prevention | Economy and trade | Public health | Environment | Society |
| | | Mean score | Mean score | Mean score | Mean score | Mean score | Mean score | Mean score |
| | | Rank | Rank | Rank | Rank | Rank | Rank | Rank |
| Swine influenza | 3 | 83.45 25 | 62.51 24 | 64.39 25 | 79.70 24 | 61.23 21* | 74.46 27 | 74.97 26 |
| Toxocariasis | 1 | 107.17 5 | 77.48 6 | 80.65 5 | 104.72 4 | 68.59 14* | 102.02 5 | 102.38 5 |
| Bovine tuberculosis | 1 | 106.05 6 | 83.47 2* | 68.92 22* | 101.16 6 | 74.85 4 | 100.78 6 | 101.05 6 |
| Bivariate Spearman rank correlation between the complete model and reduced models (Rho correlation coefficient) | | 0.866 | 0.609 | 0.993 | 0.675 | 0.983 | 0.980 | |
| P-value | | < 0.05 | < 0.05 | < 0.05 | < 0.05 | < 0.05 | < 0.05 | |

Cluster of zoonoses. 1: Very high importance, 2: High importance, 3: Moderate importance,

*Ranking changed by three or more positions.

diseases, eight animal diseases (five viral, two bacterial, and one parasitic diseases) belonged to the cluster of very high importance that is brucellosis, foot and mouth disease, high pathogenic avian influenza, low pathogenic avian influenza, monieziasis, Rift Valley fever, viral hepatitis E, and bovine tuberculosis. Of the nine zoonoses ranked in the top ten animal diseases, two (brucellosis and bovine tuberculosis) were common to both disease prioritization processes. Clusters of high importance and moderate importance for livestock disease prioritization were made of 18 and 14 diseases respectively (Fig 2).

For the prioritization of zoonoses the relative importance of the 6 domains varied depending on the disease. However, with respect to all the seven domains for the 27 diseases, the public health domain obtained the highest scores, being ranked first 15 times and never ranked at the three bottom positions (4th, 5th, 6th) (Fig 3). In addition, for the same domain the ranking of diseases changed in the corresponding reduced model for three or more positions 19 times (Table 6). Contrarily, the economy and trade domain was ranked 27 times at the three bottom positions and never at the top three ones (1st, 2nd, 3rd). Additionally, the ranking position of diseases shifted for more than three positions only once for the corresponding reduced model (Table 6).

For the prioritization of animal diseases, the relative importance of the seven domains also varied depending on disease. For the 40 selected diseases, the domain "control and prevention" was ranked 25 times first and never seventh. Other domains ranked first were "public health" with nine occurrences and "economy and trade" six times. "Economy and trade" was second 20 times and never seventh out of the seven domains. "Epidemiology–knowledge" was ranked third 16 times, and "livestock production systems" was mainly ranked at the fourth, fifth and sixth positions. The domain "environment" was mainly ranked fifth and sixth. The domain "society" was mainly ranked sixth and seventh (Fig 3). The rank of diseases changed for more than three positions 34, 29, 27 times for public health, economy and trade, control, and prevention domains respectively. Ranking of diseases changed for more than three positions only twice for society reduced model (Table 7). All the reduced models showed an excellent correlation with the complete model apart from the reduced model "public health" for the prioritization of animal diseases.

**Table 7. Ranking and two-step cluster analysis of 40 animal diseases according to the completed and reduced domain models.**

| Disease | Cluster | Reduced domain | | | | | | | |
|---|---|---|---|---|---|---|---|---|---|
| | | Complete model | Epidemiology and infectiology | Control and prevention | Economy and trade | Public health | Environment | Livestock production system | Society |
| | | Mean score | Mean score | Mean score | Mean score | Mean score | Mean score | Mean score | Mean score |
| | | Rank | Rank | Rank | Rank | Rank | Rank | Rank | Rank |
| Anaplamosis | 2 | 124.16 10 | 95.73 23* | 98.51 2* | 81.49 26* | 124.16 3* | 111.89 11 | 115.64 11 | 117.54 12 |
| Anthrax | 2 | 117.52 16 | 101.44 15 | 82.30 19* | 102.13 6* | 91.95 26* | 106.44 18 | 105.27 20* | 115.63 16 |
| ASF | 3 | 94.05 36 | 78.01 37 | 69.76 24* | 70.27 35 | 94.05 25* | 78.13 38 | 82.51 37 | 91.56 36 |
| Babesiosis | 3 | 99.87 34 | 75.62 38 | 66.21 27* | 70.01 36 | 99.87 19* | 92.00 31* | 97.67 26* | 97.84 33 |
| Brucellosis | 1 | 134.95 1 | 120.67 1 | 95.19 5* | 100.53 9* | 103.19 15* | 126.62 1 | 131.08 1 | 132.43 1 |
| BVD | 3 | 90.87 38 | 79.03 36 | 58.77 38 | 67.93 37 | 90.87 28* | 74.00 39 | 85.50 35* | 89.11 37 |
| Campylobacteriosis | 2 | 113.15 22 | 98.48 19* | 81.35 20 | 88.24 16* | 94.56 23 | 102.16 23 | 104.79 21 | 109.30 22 |
| CBPP | 2 | 107.77 26 | 89.98 28 | 70.33 23* | 75.43 32* | 107.77 11* | 101.57 24 | 96.16 28 | 105.39 26 |
| CCPP | 3 | 89.01 39 | 72.88 39 | 58.12 39 | 72.89 33* | 89.01 30* | 78.26 37 | 75.58 39 | 87.34 39 |
| Coccidiosis | 3 | 100.09 33 | 87.22 31 | 62.79 34 | 76.66 31 | 100.09 17* | 91.66 32 | 84.79 36* | 97.32 34 |
| Colibaccilosis | 2 | 116.87 17 | 99.42 18 | 65.33 29* | 88.94 15 | 113.48 9* | 111.95 10* | 106.46 18 | 115.67 14* |
| COVID-19 | 2 | 119.72 13 | 106.66 11 | 90.76 10* | 101.15 7* | 81.83 36* | 106.45 17* | 116.15 10* | 115.31 17* |
| CSF | 3 | 101.90 32 | 89.89 30 | 86.61 12* | 56.57 39* | 101.90 16* | 87.81 33 | 90.22 31 | 98.42 32 |
| Cysticercosis | 2 | 103.87 31 | 92.29 25* | 85.04 13* | 77.81 28* | 71.79 39* | 97.07 28* | 96.71 27* | 102.49 29 |
| Dermatophylosis | 2 | 125.75 8 | 116.34 4* | 65.81 28* | 95.18 14* | 125.75 2* | 111.77 12* | 116.86 8 | 122.76 8 |
| Fasciolosis | 2 | 116.23 18 | 106.55 12* | 64.52 32* | 84.67 22* | 116.23 6* | 101.34 25* | 109.64 13* | 114.42 18 |
| FMD | 1 | 132.10 2 | 117.88 3 | 94.70 6* | 80.11 27* | 127.97 1 | 122.68 3 | 118.69 7* | 130.60 2 |
| Heart water | 3 | 91.83 37 | 86.33 33* | 61.59 35 | 60.66 38 | 91.83 27* | 80.37 36 | 82.50 38 | 87.69 38 |
| Hokovirosis | 3 | 106.14 28 | 85.84 34* | 60.63 36* | 95.29 13* | 97.74 21* | 95.94 30 | 95.69 30 | 104.09 27 |
| HPAI | 1 | 124.63 9 | 110.26 8 | 110.26 8 | 107.08 4* | 82.26 35* | 116.12 6* | 116.81 9 | 121.78 9 |
| IBD | 3 | 104.48 29 | 92.38 24* | 68.51 25* | 86.48 19* | 104.48 14* | 85.79 34* | 87.10 34* | 102.14 30 |
| IBR | 3 | 106.94 27 | 87.10 32* | 65.32 30* | 82.35 24* | 106.94 13* | 96.51 29 | 99.77 25 | 103.64 28 |
| Infectious endometritis | 3 | 70.77 40 | 61.26 40 | 58.79 37* | 36.90 40 | 70.77 40 | 66.11 40 | 61.87 40 | 68.95 40 |
| Infectious mastitis | 3 | 104.48 30 | 91.36 27 | 89.38 11* | 81.57 25* | 75.61 37* | 97.52 26* | 90.04 32 | 101.37 31 |

*(Continued)*

**Table 7.** (Continued)

| Disease | Cluster | Complete model | Epidemiology and infectiology | Control and prevention | Economy and trade | Public health | Environment | Livestock production system | Society |
|---|---|---|---|---|---|---|---|---|---|
| | | | | | | | | Reduced domain | |
| | | Mean score | Mean score | Mean score | Mean score | Mean score | Mean score | Mean score | Mean score |
| | | Rank | Rank | Rank | Rank | Rank | Rank | Rank | Rank |
| Leptospirosis | 2 | 114.75 19 | 101.26 17 | 82.48 18 | 101.05 8* | 75.44 38* | 108.52 15* | 107.47 17 | 112.28 21 |
| LPAI | 1 | 128.80 7 | 114.48 6 | 84.34 15* | 108.88 3* | 99.88 18* | 115.24 7 | 123.83 4 | 126.17 7 |
| Lumpy skin disease | 2 | 118.59 15 | 91.86 26* | 84.30 16 | 83.76 23* | 116.63 5* | 110.68 14 | 108.69 15 | 115.64 15 |
| Monieziasis | 1 | 129.35 5 | 119.19 2* | 91.28 9* | 100.37 10* | 99.74 20* | 118.59 5 | 120.38 6 | 126.55 6 |
| Nematodiasis | 2 | 120.25 12 | 107.16 10 | 84.83 14 | 88.12 17* | 107.03 12 | 106.01 19* | 110.55 12 | 117.78 11 |
| Newcastle disease | 2 | 114.64 20 | 101.96 14* | 65.30 31* | 77.67 29* | 114.64 7* | 107.77 16* | 107.73 16* | 112.77 20 |
| Paramphistomosis | 3 | 98.93 35 | 84.71 35 | 70.66 22* | 72.21 34 | 96.08 22* | 83.95 35 | 89.17 33 | 96.79 35 |
| Pasteurellosis | 2 | 118.63 14 | 98.12 20* | 63.57 33* | 98.25 12 | 118.62 4* | 110.84 13 | 106.25 19* | 116.07 13 |
| PPR | 2 | 114.31 21 | 101.36 16* | 57.80 40* | 85.25 21 | 114.31 8* | 105.07 20 | 109.12 14* | 112.94 19 |
| Q-fever | 2 | 108.05 25 | 97.35 21* | 75.02 21* | 87.12 18* | 84.73 34* | 97.25 27 | 100.94 24 | 105.88 25 |
| RVF | 1 | 130.74 3 | 110.07 9* | 96.51 4 | 110.69 2 | 88.42 31* | 123.51 2 | 125.91 2 | 129.32 3 |
| Salmonellosis | 2 | 120.75 11 | 106.38 13 | 97.88 3* | 105.51 5* | 87.96 32* | 112.57 8* | 95.78 29* | 118.41 10 |
| Scabies | 2 | 111.26 23 | 95.96 22 | 83.04 17* | 85.71 20* | 86.21 33* | 103.55 22 | 104.39 22 | 108.71 23 |
| Trypanosomiasis | 2 | 110.09 24 | 89.92 29* | 67.95 26 | 77.25 30* | 110.09 10* | 104.24 21* | 102.84 23 | 108.24 24 |
| Bovine tuberculosis | 1 | 129.23 6 | 115.49 5 | 106.14 1* | 99.20 11* | 94.51 24* | 112.35 9* | 121.01 5 | 126.68 5 |
| Viral Hepatitis E | 1 | 129.61 4 | 113.72 7* | 92.05 8* | 111.45 1* | 89.33 29* | 119.49 4 | 124.53 3 | 127.10 4 |
| Bivariate Spearman rank correlation betwee the complete model and reduced models (Rho correlation coefficient) | | 0.930 | 0.671 | 0.794 | 0.277 | 0.968 | 0.938 | 0.995 | |
| P-value | | < 0.05 | < 0.05 | < 0.05 | = 0.084 | < 0.05 | < 0.05 | < 0.05 | |

Cluster of animal diseases. 1: Very high importance, 2: High importance, 3: Moderate importance.

*Ranking changed by three positions.

ASF: African swine fever; BVD: Bovine viral disease; CBPP: Contagious bovine pleuropneumonia; CCPP: Contagious caprine pleuropneumonia; CSF: Classical swine fever; FMD: Foot and mouth disease; HPAI: High pathogenic avian influenza; IBD; infectious bursal disease, IBR: Infectious bovine rhinotracheitis, LPAI: Low pathogenic avian influenza; PPR: Small ruminant plague, RVF: Rift Valley fever.

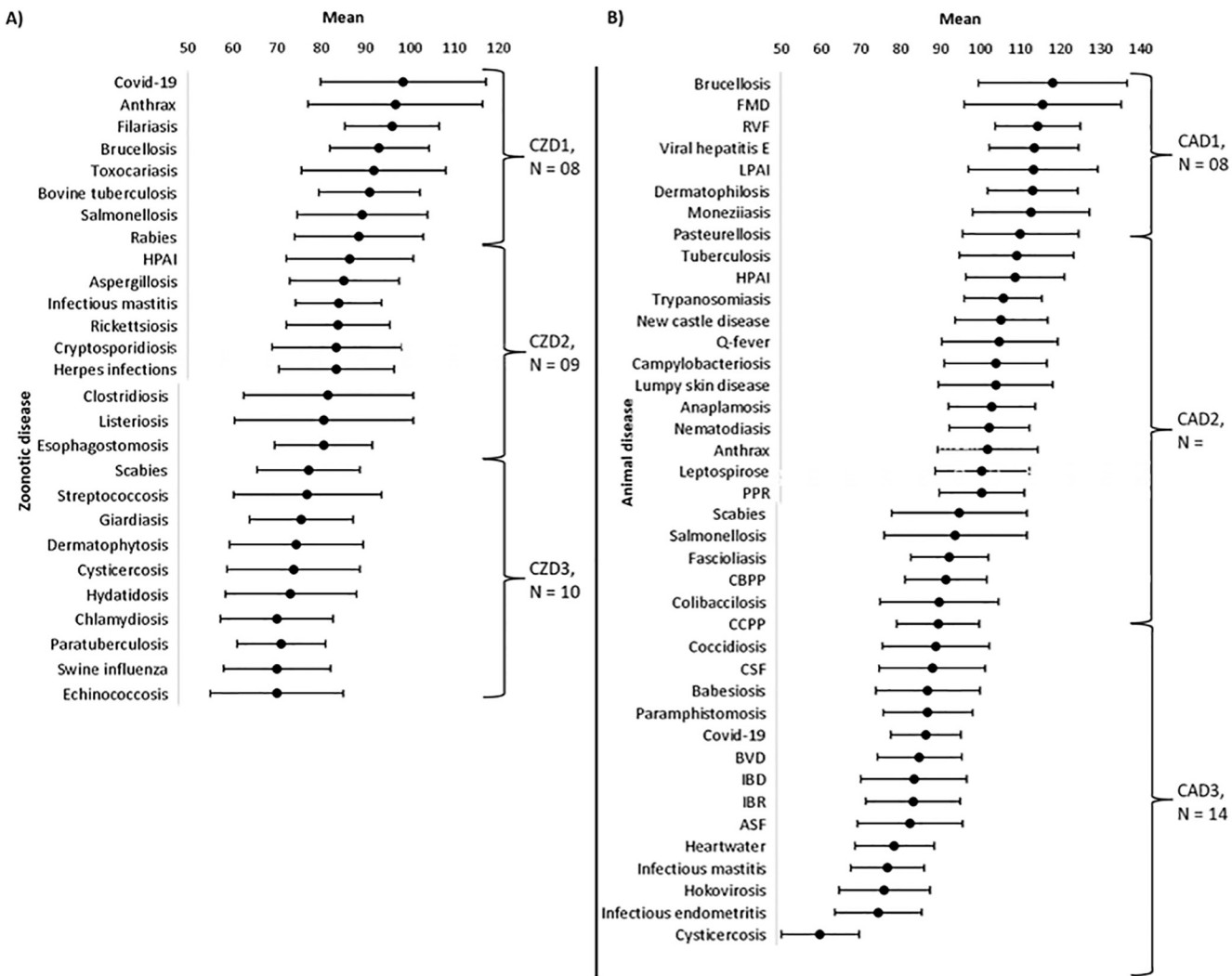

**Fig 2. Mean score and standard deviation of zoonoses and animal diseases prioritized.** Mean scores and standard deviations are mentioned. For each prioritization process, three clusters marked by brackets were identified by two step cluster analysis for the prioritization of zoonoses (A) and animal diseases (B) respectively. For cluster, CZD: Cluster of zoonotic diseases; CAD: Cluster of animal diseases. 1: Very high importance, 2: High importance, 3: Moderate importance, 4: Low importance. ASF: African swine fever; BVD: Bovine viral disease; CBPP: Contagious bovine pleuropneumonia; CCPP: Contagious caprine pleuropneumonia; CSF: Classical swine fever; FMD: Foot and mouth disease; HPAI: High pathogenic avian influenza; IBD; infectious bursal disease, IBR: Infectious bovine rhinotracheitis, LPHAI: Low pathogenic avian influenza; PPR: Small ruminant plague.

The ranking movements of the top five diseases for zoonosis and livestock disease prioritizations changed according to each disease. For the prioritization of zoonoses, upon exclusion of "public health" domain, COVID-19 was transferred from first to tenth rank, thus underlining the strong influence of the public health domain on this specific disease. Anthrax ranking changed significantly for the reduced model "control and prevention" from second to 14[th] rank. Brucellosis was also strongly influenced, moving from fourth to first rank, in the reduced model without "public health domain". Concerning the prioritization of animal diseases, upon exclusion of "control and prevention", "economy and trade", "public health domains, three and four diseases out the five diseases had their ranks changed three times or more (Fig 4). In addition, the ranges of the mean scores were higher for zoonoses (apart from rabies and aspergillosis) than animal diseases (S1 and S2 Figs).

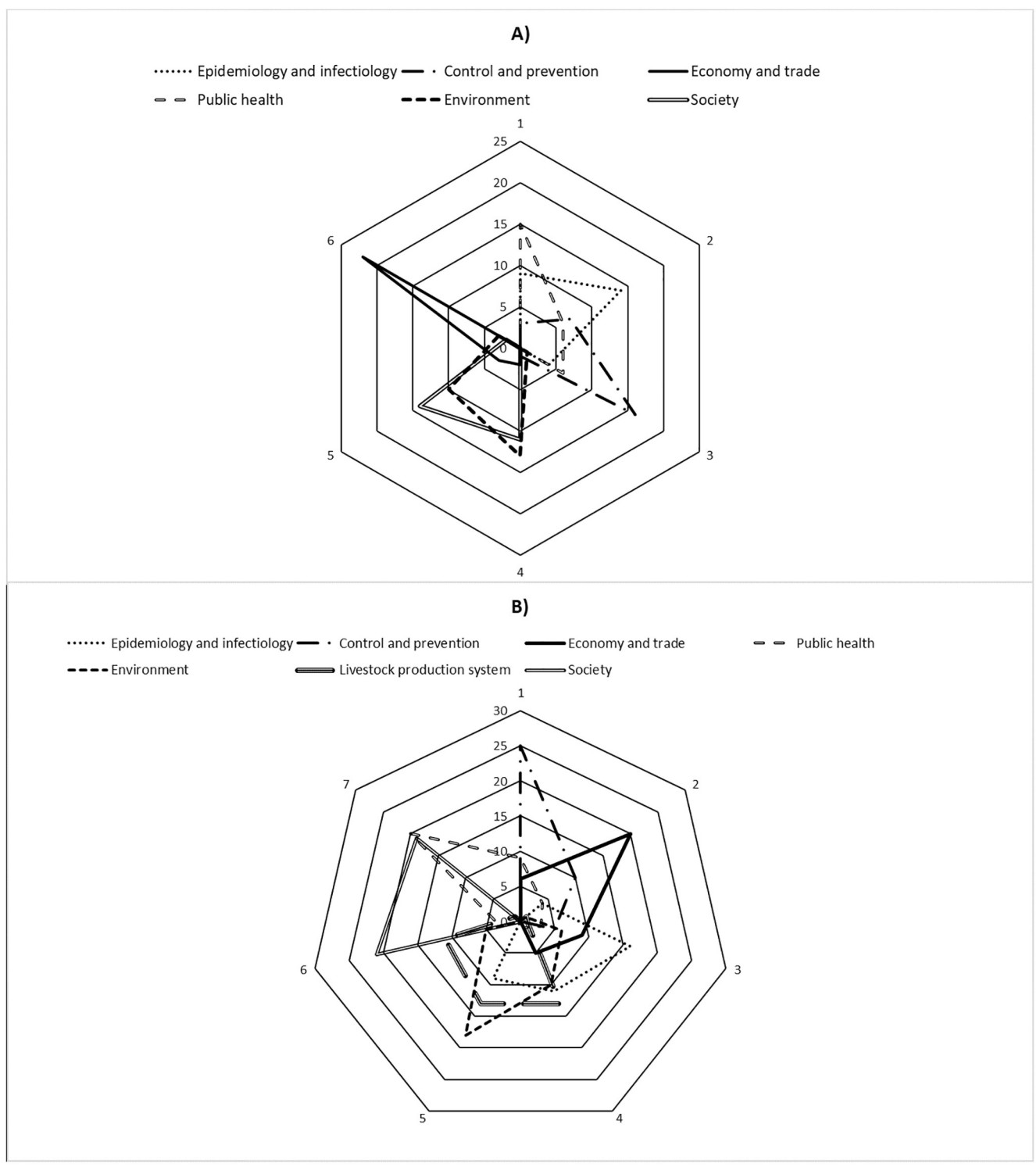

**Fig 3.** Frequency of rank using spider webs: From 1 to 6 for zoonoses (A) and 1 to 7 for animal diseases (B). The line inside the spider web displays the relative performance (count) of each domain for the different ranks (represented by 1–7).

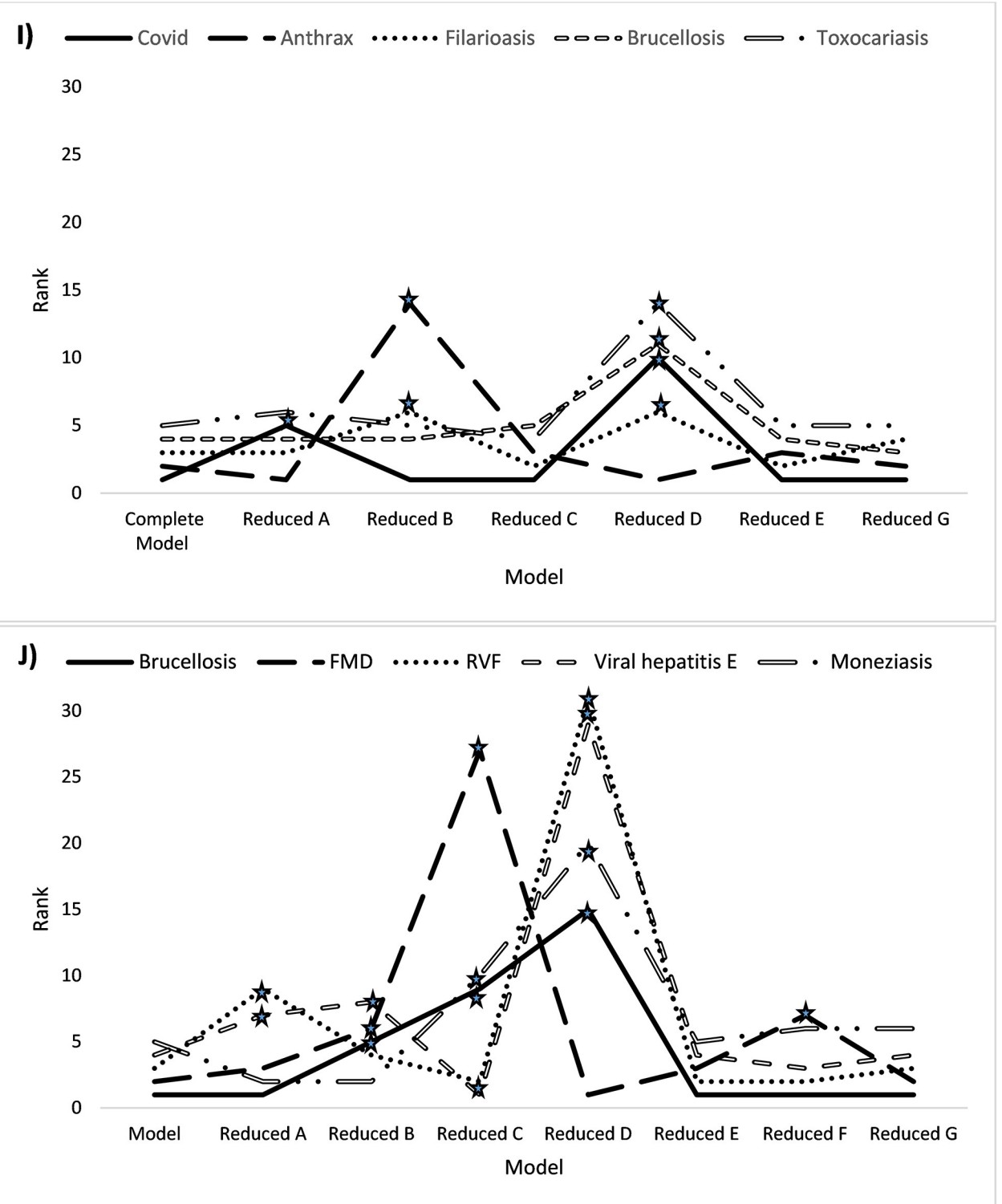

**Fig 4. Sensitivity analysis for the five zoonoses (I) and five animal diseases (J) with highest mean scores.** The graph illustrates the trends upward or downward in the ranking. *Ranking changed by three positions or more. (A) Epidemiology and infectiology; (B) Control and prevention; (C) Economy and trade; (D) Public health; (E) Environment; (F) Livestock production system and (G) Society. FMD, foot-and-mouth disease; RVF, Rift Valley fever. Each reduced model derived from the complete model with the exclusion of one domain of criteria (i.e. A, B, C, D, E, F or G).

## 5. Discussion

MCDA methods are objective tools for informed decision-making processes in animal and human health, since they rely on varying scales and perceptions of impacts using a structured, transparent approach, and they aim to give consistent, reproducible results [12]. To the best of our knowledge, this is the first time MCDA has been used in the context of endemic zoonoses and animal diseases at local (regional) level in SSA or LMICs. In addition, in this study, the MCDA used by other authors [16,17] has been associated to a multidimensional analysis (CATPCA) for the reduction of dimensions or criteria, and two-step cluster analysis used in place of regression tree analysis. Those analyses have been applied in other contexts to reduce the number of variables and focus on discriminating classification variables from surveys aiming at identifying and classifying livestock production systems in several countries [18,23,29] and can be applied in any multivariate decision-making setting. Indeed, the weight distribution in the study was left to the participating experts. The direct scoring method used in the study was like the one used in previous prioritization processes [16,17,30–32] because it associated numerical scales, rather than non-informative ad-hoc scales. By using CATPCA (allowing the identification of uncorrelated criteria with high variance accounted for used in disease classification) the mutual preferential independence (one of the main assumptions for MCDA such as the Weighed Sum Model (WSM), the Weighed Product Model (WPM) [33] and the Analytic Hierarchy Process (AHP) [34]) ensuring that the preference for any one criterion did not depend on the values of the other criteria was achieved. Thus, the necessary condition of incorporating criteria interactions when dealing with dependent criteria in more complex MCDA was irrelevant [35].

There are several reasons to prioritize zoonoses and animal diseases in parallel. Animal diseases in general include both zoonoses and non-zoonotic animal diseases. Zoonoses can also affect humans and non-livestock animals (wildlife or pets). In addition, prioritization of zoonoses must be an integrated and open process allowing a multisector or One Health approach to enhance coordination, collaboration and communication between human, animal, and environment health sectors [8,9,13]. Prioritization of animal diseases requires the involvement of professionals and stakeholders of livestock production and health at different levels. The experts used in this study are balanced accordingly to take into consideration those characteristics. The number of participants to disease prioritization processes varies widely from one to several dozens and their roles during the process can also vary from scoring, voting, facilitating, and observing [8,17]. In this study, participants are distinguished into two categories: criteria experts and disease experts. An expert can be part of both groups to ensure good linkages between criteria, disease scoring and weighing steps by mixing experts participating in both processes to those involved in only one.

By applying inclusion and exclusion criteria two lists were generated: the list of 27 zoonoses (11 bacterial, 11 parasitic, and five viral diseases) and the list of 40 animal diseases (15 bacterial, ten parasitic and 15 viral diseases). Previous zoonotic disease prioritization processes carried out in SSA used lists with similar number of diseases (for the prioritization of zoonoses using the CDC Atlanta One Health Zoonotic Diseases Prioritization) or shorter lists of diseases (zoonoses and animal diseases) with the OIE Phylum prioritization tool [9]. Regardless of the prioritization scales and depending on the prioritization objectives used for disease inclusion and exclusion criteria, the number of diseases also varied in other prioritization settings in Europe and other continents. Thus, 83 diseases were selected for prioritization as part of a wildlife disease surveillance strategy in New Zealand [36], 103 animal diseases listed according to their consequences on animal health and public health in France [30], 51 foodborne zoonoses in Belgium [31], 174 zoonoses to support the development of national surveillance systems for

emerging zoonoses in Netherlands [32], 100 diseases of food-producing animals and zoonoses in Belgium for prioritization purpose [16], 14 diseases to establish priorities for emerging or re-emerging infectious diseases associated with climate change in Canada [37] and 29 diseases to prioritize animal diseases based on factors of emergence and re-emergence in Belgium [17]. Prioritization of animal diseases often covers zoonoses. Thus, in this study nine zoonoses were common to both parallel processes. Also depending on the prioritization objectives, a list of diseases could be limited to a type of diseases such as infectious diseases having epidemic or pandemic potential such as viral diseases [17].

The number of domains and criteria used for prioritization of zoonoses, and animal diseases are also subject to variation depending on the prioritization objectives and the context of the process. If some domains (disease impact domain) are almost always present other are likely to be less frequent [8,9]. In this study by using CATPCA from seven domains and 70 criteria, 17 and 19 criteria of six and seven domains were identified for the prioritization of zoonoses and animal diseases respectively. The seven proposed domains were epidemiology and infectiology, control and prevention, economy and trade, public health, environment and society, livestock production system. The latter was missing for the prioritization of zoonoses. The reason behind the systematic cancellation of the impact on livestock production system domain for the prioritization of zoonoses may be that the emphasis was on the zoonotic feature of diseases and not on diseases of food producing animals. The fact that more than half of the selected criteria were common to both processes showed a certain degree of convergence of expert's perception of criteria relevance and importance for both processes as already highlighted in 16 African countries for disease prioritization at national level [9]. The number of criteria used for disease prioritization also varied considerably from less than ten [31,32,36,38] to more than 40 [16,17,37]. Previous prioritization of zoonoses at national level used five criteria (similar to domains in the present study) including: 1) the state of the disease in humans, domestic animals or wild animals in Cameroon; 2) mortality, morbidity and disability in humans; 3) the potential of rapid spread among animals and humans; 4) social, environmental and economic impacts and finally 5) capacity for the detection, prevention and control of zoonotic diseases in the country [39]. Similar domains had been used in the prioritization of zoonoses in Belgium [16] and Canada (Quebec) [40]. Bianchini et al., [17] prioritized animal diseases based on drivers of emergence and reemergence rather than the impact of disease using eight domains and 50 criteria.

For zoonotic disease prioritization the classification ranked COVID-19 (SARS-CoV-2) at the top of the list of priority diseases. It was ranked first (the highest mean score) in five out of seven models certainly because of its highly transmissible nature, control-prevention challenges, high human mortality rate, and pandemic nature [41] which also affected economy and trade worldwide. SARS-CoV-1 a similar disease to COVID-19 was already ranked fifth among priority zoonotic diseases in Cote d'Ivoire [42] even though it did not result in a pandemic like COVID-19. Thus, the ongoing COVID-19 pandemic, associated impacts and uncertainties may have contributed to making it the first priority zoonosis [43]. Beside COVID-19, the other four top zoonoses were anthrax, brucellosis, filariasis and toxocariasis.

For animal diseases the top seven diseases were: brucellosis, foot and mouth disease, Rift Valley fever, viral hepatitis E, pathogenic avian influenza (LPAI and HPAI), monieziasis and bovine tuberculosis. The priority diseases at the national level were twelve: rinderpest (eradicated but prioritized to keep the country disease free status), foot-and-mouth disease, contagious bovine pleuropneumonia, African swine fever, peste des petits ruminants, Newcastle disease, HPAI, rabies, bovine tuberculosis, Ebola, brucellosis, and anthrax [44]. Indeed, it was in line with the list of priority diseases at the national level for diseases such as foot-and-mouth disease, pathogenic avian influenzas (HPAI), bovine tuberculosis and brucellosis. The priority

diseases identified in this study that are not included on the national list were Rift Valley fever, viral hepatitis E and monieziasis. Apart from viral hepatitis E detected in pigs in Cameroon [45,46], these diseases are diseases subject to notification to the OIE [47]. Anthrax, bovine tuberculosis, and brucellosis are among the thirteen priority zoonoses in Cameroon [48] and have been listed as top priority zoonoses in several other zoonoses prioritization processes in SSA [9].

Three clusters were identified for each prioritization process. Two third of the zoonoses were grouped together in the first two clusters, and for animal diseases almost half of the diseases were in the second cluster. All clusters were composed of different disease types (bacterial, parasites, viral diseases) such as bovine tuberculosis, brucellosis, trypanosomiasis, helminthiasis, tick borne diseases and foot and mouth diseases that had been well described in the country [7].

For the prioritization of zoonoses, domain influence on disease classification was more marked in public health and control-prevention reduced models. In addition to those domains, economy and trade also significantly influenced animal disease classification highlighting the importance of that domain for livestock production and food of animal origin value chain contrarily to the prioritization of zoonoses where it was the least influencing domain. These results were like those of prioritization processes carried out using the CDC One Health zoonotic Diseases Prioritization tool and the OIE phylum tool in SSA showing that it was first the criteria linked to the impact on public health which were most often cited as important and among them the direct consequences of zoonoses in terms of mortality and morbidity [9].

Sensitivity analysis performed on experts showed that disease classification was influenced by experts. When an expert group was removed, the disease rank changed by at least three positions (up or down) in at least two expert reduced models. This meant that for the prioritization of zoonoses, expert opinions varied for the disease they had in common due to the mixed background of members of each expert group made up of people from various sectors such as veterinarians, human health professionals and wildlife/environment health workers. Indeed, the scoring of each criterion for zoonoses could vary in importance due to expertise biases consequently changing according to the field of expertise (13). This was different from the prioritization of animal diseases where sensitivity analysis carried out on expert groups showed that reduced models yielded similar ranking to complete model with a high positive correlation between the complete model and reduced models (S5 and S6 Tables). Thus, there was consistency and agreement in scoring and classification of diseases by experts for the prioritization of animal diseases. The present method could be applied at the local level in the context of SSA for the prioritization of zoonoses and animal diseases to enhance preparedness, investigation, and response to zoonoses and animal health events.

The limitation of the present study was mainly the fact that the evaluation of criterion/disease relevance (score) and importance (intra/interdomain weight) might not be known by experts with certainty resulting from a distinct "lack" of complete knowledge (S1 and S2 Figs). The so-called uncertainty deriving from many sources and possessing various forms grouped within "external" uncertainty related to the environmental conditions not under the control of experts [49] or "internal" uncertainty, i.e., uncertainties about expert preferences and problem identification (e.g., problem structuring methods [50]), vagueness of information (e.g., rough set theory [51]), and imprecise judgements. However, the latter was addressed using sensitivity analysis for experts (S5 and S6 Tables). Such process can also be complemented or associated to a validation exercise involving horizontal and vertical cross-sectoral relevant stakeholders aiming at consensual adjustments of priority diseases [9].

## 6. Conclusions

Countries in SSA are strongly constrained by several zoonoses and livestock diseases that have various economic and public health impacts. In this study endemic zoonoses and animal diseases circulating in Cameroon and neighbouring countries were prioritized at local level. However, despite the prioritization scale, this exercise can be carried out in different regions to account for specificities related to different agro-ecological zones shaped by climatic variations and production systems. Indeed, the focus of the questionnaire was to prioritize diseases according to their impact locally. This was achieved in a parallel way both for zoonoses and animal diseases aiming at having a sound and balanced representativeness of all stakeholders at all levels. The number of identified criteria was reduced using CATPCA to focus on the most discriminating domains and criteria for prioritization. Sensitivity analysis of experts showed both low and high correlation among the ranking of models for the prioritization of zoonoses and animal diseases respectively. Sensitivity analysis of domains yielded mainly high correlation highlighting the importance of validating each generated model while identifying domains highly influencing disease ranking. Given the structured nature of criteria and the scoring system, this tool is highly reproducible and transparent in a standard way. In addition to the outcomes of this process the methodology contextualised in this study can be used in different settings. Thus, in SSA the operationalization of prioritization of zoonoses and animal diseases at the regional, country, local and farm levels can be improved by involving all stakeholders particularly at the local level through a standardized prioritization methods such as MCDA whereby the disease ranking yielded can be used to design tailored site-specific actions or operational plan to control both zoonoses and animal diseases.

## Supporting information

**S1 Fig. Graph showing the mean, median scores and the range of the scores among the experts per zoonosis.**
(PDF)

**S2 Fig. Graph showing the mean, median scores and the range of the scores among the experts per animal disease.**
(PDF)

**S1 Table. Criteria experts involved in the prioritization of zoonoses.**
(PDF)

**S2 Table. Criteria experts involved in the prioritization of animal diseases.**
(PDF)

**S3 Table. Disease experts involved in the prioritization of zoonoses.**
(PDF)

**S4 Table. Disease experts involved in the prioritization of animal diseases.**
(PDF)

**S5 Table. Ranking and two-step cluster analysis of 36 zoonoses according to the completed and reduced expert models.**
(PDF)

**S6 Table. Ranking and two-step cluster analysis of 40 animal diseases according to the completed and reduced expert models.**
(PDF)

**S1 Data.**

(XLSX)

**S2 Data.**

(XLSX)

**S3 Data.**

(XLSX)

**S4 Data.**

(XLSX)

## Acknowledgments

The authors thank all experts who participated to this study. The list of experts can be found on S1 to S4 Tables.

## Author Contributions

**Conceptualization:** Serge Eugene Mpouam, Jean Pierre Kilekoung Mingoas, Claude Saegerman.

**Data curation:** Serge Eugene Mpouam.

**Formal analysis:** Serge Eugene Mpouam.

**Funding acquisition:** Serge Eugene Mpouam, Jean Pierre Kilekoung Mingoas, Claude Saegerman.

**Investigation:** Serge Eugene Mpouam, Dalida Ikoum, Limane Hadja.

**Methodology:** Serge Eugene Mpouam, Claude Saegerman.

**Project administration:** Serge Eugene Mpouam, Jean Pierre Kilekoung Mingoas, Claude Saegerman.

**Resources:** Serge Eugene Mpouam, Jean Pierre Kilekoung Mingoas, Claude Saegerman.

**Software:** Serge Eugene Mpouam, Claude Saegerman.

**Supervision:** Serge Eugene Mpouam, Jean Pierre Kilekoung Mingoas, Claude Saegerman.

**Validation:** Serge Eugene Mpouam, Claude Saegerman.

**Visualization:** Serge Eugene Mpouam, Claude Saegerman.

**Writing – original draft:** Serge Eugene Mpouam.

**Writing – review & editing:** Serge Eugene Mpouam, Dalida Ikoum, Limane Hadja, Jean Pierre Kilekoung Mingoas, Claude Saegerman.

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
