## [Decision Letter · Decision Letter 0]

18 Sep 2023

PONE-D-23-22715

Parallel multi-criteria decision analysis for sub-national prioritization of zoonoses and animal diseases in Africa: the case of Cameroon

PLOS ONE

Dear Dr. Saegerman,

Thank you for submitting your manuscript to PLOS ONE. After careful consideration, we feel that it has merit but does not fully meet PLOS ONE’s publication criteria as it currently stands. Therefore, we invite you to submit a revised version of the manuscript that addresses the points raised during the review process.

We look forward to receiving your revised manuscript.

Kind regards,

Daniel Oladimeji Oluwayelu, D.V.M., M.Sc., Ph.D.

Academic Editor

PLOS ONE

“This work is funded through the Innovative Methods and Metrics for Agriculture and Nutrition Action (IMMANA) programme, led by the London School of Hygiene & Tropical Medicine (LSHTM). IMMANA is co-funded with UK Aid from the UK government and by the Bill & Melinda Gates Foundation INV-002962 / OPP1211308. Under the grant conditions of the Founda-tion, a Creative Commons Attribution 4.0 Generic License has already been assigned to the Au-thor Accepted Manuscript version that might arise from this submission.”

Reviewers' comments:

Reviewer's Responses to Questions

**Comments to the Author**

1. Is the manuscript technically sound, and do the data support the conclusions?

Reviewer #1: Yes

Reviewer #2: Yes

2. Has the statistical analysis been performed appropriately and rigorously? 

Reviewer #1: Yes

Reviewer #2: Yes

3. Have the authors made all data underlying the findings in their manuscript fully available?

Reviewer #1: Yes

Reviewer #2: Yes

4. Is the manuscript presented in an intelligible fashion and written in standard English?

Reviewer #1: Yes

Reviewer #2: Yes

5. Review Comments to the Author

Reviewer #1: The data presents an important topic on disease prioritization and the merits of doing the same at sub-national level. The paper is well-written, featuring a clear and concise presentation of the methods, results, and a thorough discussion. The data presents an important topic on disease prioritization and the merits of doing the same at sub-national level.

Reviewer #2: Please see my comments within your submission. Overall this work is worthy of publication, but there are inconsistencies in the style of writing and formatting. I have highlighted many instances of numbers that should have been written that were not and vice versa. Please review and correct throughout the paper.

Lines 266-295 and 331-388 are especially awkward and I highly recommend revisions to clarify what is being said.

6. PLOS authors have the option to publish the peer review history of their article (what does this mean?). If published, this will include your full peer review and any attached files.

Reviewer #1: No

Reviewer #2: No

---

## [Author Response · Author response to Decision Letter 0]

4 Oct 2023

All comments of the reviewers were taken into account and a note of revision was added.

---

## [Editor Report · Decision Letter 1]

9 Oct 2023

PONE-D-23-22715R1Parallel multi-criteria decision analysis for sub-national prioritization of zoonoses and animal diseases in Africa: the case of CameroonPLOS ONE

Dear Dr. Saegerman,

Thank you for submitting your manuscript to PLOS ONE. After careful consideration, we feel that it has merit but does not fully meet PLOS ONE’s publication criteria as it currently stands. Therefore, we invite you to submit a revised version of the manuscript that addresses the points raised during the review process.

We look forward to receiving your revised manuscript.

Kind regards,

Daniel Oladimeji Oluwayelu, D.V.M., M.Sc., Ph.D.

Academic Editor

PLOS ONE

Journal Requirements:

Additional Editor Comments:

1. Authors should correct all typographical and other errors in the manuscript e.g., SSA (Line 65) should be written in full at first use (Line 49); Pubmed should be changed to PubMed, Plos One to PLOS ONE, African journal online to African Journals Online (Line 118-119); valley to Valley (Line 263 and 311), and african to African (Line 309); Table 4 - New Castle to Newcastle (No. 28); valley to Valley (No. 29).

2. Response to Comment 6: The phrase highlighted in yellow (line 172-174) by the reviewer was not deleted. Authors should see the phrase “...set, finally resulted in the identification of the most discriminating criteria that were therefore used as disease prioritization criteria.” (Line 173-175) in the revised manuscript and make necessary corrections.

---

## [Author Response · Author response to Decision Letter 1]

25 Nov 2023

RESPONSE TO REVIEWERS

Dear Editor,

Thank you for your email about our manuscript [PONE-D-23-22715]- [EMID:7af7abfb763bdaab]. We appreciate reviewers and editor’s comments to improve the quality of our paper in order to meet PLOS ONE’s publication criteria.

Journal requirements. 

Reference list reviewed. The list is complete and correct

Comments and responses

Comment 1. Authors should correct all typographical and other errors in the manuscript e.g., SSA (Line 65) should be written in full at first use (Line 49); Pubmed should be changed to PubMed, Plos One to PLOS ONE, African journal online to African Journals Online (Line 118-119); valley to Valley (Line 263 and 311), and african to African (Line 309); Table 4 - New Castle to Newcastle (No. 28); valley to Valley (No. 29).

Response 1. Changes were made as requested.

Comment 2. Response to Comment 6: The phrase highlighted in yellow (line 172-174) by the reviewer was not deleted. Authors should see the phrase “...set, finally resulted in the identification of the most discriminating criteria that were therefore used as disease prioritization criteria.” (Line 173-175) in the revised manuscript and make necessary corrections. 

Note that now Table 4 becomes Table 5.

Response 2. The reviewer’s comment was: « Table 2 appears to be the inclusion exclusion criteria of diseases, not the prioritization criteria”. Thus, we were supposed to give the correct table number for the phrase and not delete the phrase highlighted in yellow. Therefore we deleted “(Table 2)” for the previous reviewed manuscript and now replaced with “(Table 4)” which is the appropriate table number for the phrase.

---

## [Editor Report · Decision Letter 2]

29 Nov 2023

Parallel multi-criteria decision analysis for sub-national prioritization of zoonoses and animal diseases in Africa: the case of Cameroon

PONE-D-23-22715R2

Dear Prof. Claude Saegerman, 

We’re pleased to inform you that your manuscript has been judged scientifically suitable for publication and will be formally accepted for publication once it meets all outstanding technical requirements.

Kind regards,

Daniel Oladimeji Oluwayelu, D.V.M., M.Sc., Ph.D.

Academic Editor

PLOS ONE
---

## [Editor Report · Acceptance letter]

8 Apr 2024

PONE-D-23-22715R2 

PLOS ONE

Dear Dr. Saegerman, 

I'm pleased to inform you that your manuscript has been deemed suitable for publication in PLOS ONE. Congratulations! Your manuscript is now being handed over to our production team.

Kind regards, 

on behalf of

Professor Daniel Oladimeji Oluwayelu 

Academic Editor

PLOS ONE